The interactive effect of herbivory, nutrient enrichment and mucilage on shallow rocky macroalgal communities

Caronni Sarah sarah.caronni@unipv.it 1
Calabretti Chiara 1
Citterio Sandra 1
Delaria Maria Anna 2
Gentili Rodolfo 1
Macri Giovanni 3
Montagnani Chiara 1
Navone Augusto 4
Panzalis Pieraugusto 4
Piazza Giulia 1
Ceccherelli Giulia 2
1 Department of Earth and Environmental Sciences, University of Milan—Bicocca , Milan , Italy
2 Department of Science for Nature and Environmental Resources, University of Sassari , Sassari , Italy
3 Mac Pro e Gis , Pavia , Italy
4 Marine Protected Area of Tavolara Punta Coda Cavallo , Olbia , Italy
Nelson Craig
Electronic publication date: 2019 May 13
Publication date: 2019
Volume: 7
Electronic Location ID: e6908
Received 2018 Feb 28; Accepted 2019 Apr 4
Copyright: ©2019 Caronni et al.
Copyright year: 2019
Copyright holder: Caronni et al.
License: This is an open access article distributed under the terms of the Creative Commons Attribution License, which permits unrestricted use, distribution, reproduction and adaptation in any medium and for any purpose provided that it is properly attributed. For attribution, the original author(s), title, publication source (PeerJ) and either DOI or URL of the article must be cited.
License URL: https://creativecommons.org/licenses/by/4.0/

Keywords: Multiple stressors, Mediterranean Sea, Macroalgae, Interactive effects, Mucilage

Funding: Sarah Caronni with a L’Oréal-Unesco for Women in Science Fellowship Grant This work was supported by L’Oréal-Unesco, which funded Sarah Caronni with a L’Oréal-Unesco for Women in Science Fellowship Grant. The funders had no role in study design, data collection and analysis, decision to publish, or preparation of the manuscript.

==============================
This paper focuses on the interactive short and long-term effect of three different stressors on a macroalgal assemblage. Three stressors are considered: herbivory, nutrients and mucilage. The experiment was conducted in Tavolara Punta Coda Cavallo Marine Protected Area (Mediterranean Sea) during a bloom of the benthic mucilage-producing microalga Chrysophaeum taylorii (Pelagophyceae); this microalga is recently spreading in the Mediterranean Sea. On a rocky substratum, 36 plots 20 × 20 cm in size were prepared. Factorial combinations of three experimental treatments were applied in triplicate, including three grazing levels crossed with two nutrient enrichment and two mucilage removal treatments. Significant differences were observed among treatments 8 weeks later, at the end of summer. In particular, dark filamentous algae were more abundant in all enriched plots, especially where mucilage and macroalgae had been removed; a higher percent cover of crustose coralline algae was instead observed where nutrients had been increased and no grazing pressure acted. Furthermore, the abundance of Dictyota spp. and Laurencia spp. was significantly higher in enriched mucilage-free plots where the grazing pressure was null or low. However, the effects of the treatments on the overall assemblage of the macroalgal community were not long persistent (36 weeks later). These results illustrate the capacity of a shallow-water macroalgal community to quickly recover from the simultaneous impacts of herbivory, nutrient enrichment, and mucilage.

Introduction

Marine ecosystems and especially near-shore coastal areas are typically subjected to several natural and anthropogenic abiotic and biotic stressors, which can seriously affect the structure of habitats and produce nearly irreversible shifts, leading to a significant reduction of ecosystem resistance and resilience (Adams, 2005; Claudet & Fraschetti, 2010; Guarnieri et al., 2014). A substantial research effort has therefore been done to investigate the effects of the most widespread threats to marine environment (Crain et al., 2009); anyway, understanding the complex effects of multiple stressors on marine communities still represents one of the major challenges in marine ecology (Sala et al., 2000; Zeidberg & Robinson, 2007; Guarnieri et al., 2014). Moreover, whether stressors are more harmful in combination than alone is still widely unanswered (Zeidberg & Robinson, 2007). Interactions among multiple stressors, where often the ecological effect of one stressor depends on the magnitude of another, are very common across ecosystems (Jackson et al., 2016) and several scenarios can occur when they simultaneously act: their effects can be cumulative, synergistic or antagonistic (Vinebrooke et al., 2004). Two or more stressors are defined as cumulative if their result is the mere sum of the effects of each of them (combined effect: stressor a + stressor b), while they are synergistic or antagonistic when their combined effect is respectively larger or smaller than the one expected by each of them (synergistic = combined effects are larger than adding stressor a + stressor b; antagonistic = combined effects are lower than adding stressor a + stressor b) (Crain, Kroeker & Halpern, 2008). For example, pH usually can have an antagonistic effect on heavy metal availability in water environments, as the heavy metal solubility normally increases when pH decreases (Millero et al., 2009). On the contrary, a synergistic effect has been described for Daphnia: Folt et al. (1999) observed that the reproduction rate of the crustacean increased when high temperatures and food levels were simultaneously present, while the positive effect of temperature on the population growth rate was smallest at limiting food levels (Folt et al., 1999).

In coastal ecosystems, canopy macroalgae play critical ecological roles (Koch et al., 2013), contributing significantly to total primary production and deeply affecting higher trophic levels (Graham, 2004). In fact, they provide the three-dimensional structure of marine faunal habitats and facilitate larval settlement of marine invertebrates (Rodríguez, Ojeda & Inestrosa, 1993). In the last decades, shallow temperate reefs are experiencing a dramatic reduction and loss of macroalgal habitats and their replacement by persistent barren grounds are of increasing concern (Guidetti et al., 2003; Sala et al., 2012). Several studies highlighted the importance of herbivore pressure in defining the composition and the abundance of macroalgal communities (i.e., Karez et al., 2004; Arévalo, Pinedo & Ballesteros, 2007). In particular, the abundance of herbivores (mainly sea-urchins and limpets) increases in relation to over-exploitation by large-sized predators, producing a rapid shift in dominance from canopy forming to crustose macroalgae with the creation of persistent barrens worldwide (Filbee-Dexter & Scheibling, 2014; Piazzi, Bulleri & Ceccherelli, 2016). However, not only herbivore pressure but also nutrient enrichment can alter the structure of benthic communities (Smith, Hunter & Smith, 2010) and it can determine macroalgal abundance, even if its importance is extensively debated (i.e., McGlathery, 2001; Armitage et al., 2005). Recently, it has been suggested that nutrient enrichment may mitigate over-grazing by enhancing algal growth (Boada et al., 2017).

The relationship between nutrient enrichment and microalgal abundance has been extensively described in literature (e.g., Hecky & Kilham, 1988; Anderson, Glibert & Burkholder, 2002), as nutrient enrichment is an important driver of microalgal proliferation (Smith & Schindler, 2009). There are numerous examples worldwide of harmful algal blooms linked to an increased nutrient loading and there is a growing consensus that degraded water quality caused by nutrient pollution, in particular by N compounds, contributes to the development and persistence of microalgal proliferations and, consequently, of mucilage abundance (Obernosterer & Herndl, 1995; Rinaldi et al., 1995). The presence of the mucilage corresponds to the appearance of a gelatinous material suspended in marine waters (pelagic aggregates) or covering large portions of the substratum (benthic aggregates); it is primarily produced by planktonic or benthic microalgae respectively (Mingazzini & Thake, 1995). Mucilage is a problem during blooms when microalgae proliferate rapidly, reaching very high-density, and covering large portions of substrata, as described by Schiaparelli et al. (2007) or Lugliè et al. (2008).

Detrimental effects due to mucilage proliferation can affect macroalgae and other benthic organisms and should therefore be considered an emerging threat to coastal ecosystems (Claudet & Fraschetti, 2010). In fact Devescovi & Iveša (2007) observed that primary branches of macroalgae usually show signs of necrosis after blooms (Devescovi & Iveša, 2007) and Misic, Schiaparelli & Covazzi Harriague (2011) assert that the persistence of mucilage on hard substrata can overgrow algae and cause their depletion. Finally, thick mucilage carpet that usually covers the substratum during blooms can be gradually torn off by the continuous twisting action of currents, engulfing and mechanically detaching macroalgae (Schiaparelli et al., 2007). Nevertheless, only a few papers describe the detailed effects of mucilage on macroalgal communities, and currently no information about effects of the interaction among nutrient enrichment, herbivory and mucilage on the above-mentioned communities are available in literature.

In this paper, results of a manipulative experiment to evaluate the interactive short- and long-term effect of the three above mentioned stressors (herbivory, nutrient enrichment and mucilage) on macroalgal assemblages are presented. The experiment was conducted during a bloom of the benthic mucilage-producing microalga Chrysophaeum taylorii Lewis and Bryan (Pelagophyceae), recently spreading in the Mediterranean Sea (Caronni et al., 2014; Caronni et al., 2015). C. taylorii can exude large amounts of mucilaginous material macroscopically visible when its cells density reaches values over 1,000 cells ml−1 (Caronni et al., 2014). Although it is considered a public nuisance also in its native range (Atlantic and Pacific Ocean), its blooms are infrequent there (at least in the Great Barrier Reef) and their effects are not so detrimental as those of other microalgae (e.g., Schaffelke et al., 2004). Conversely, in the Mediterranean Sea, large portions of hard rocky and sandy substrata as well as seagrasses and macroalgae can be covered by C. taylorii mucilage during summer (Lugliè et al., 2008; Caronni et al., 2011; Caronni et al., 2015). For this reason, an investigation on the effects of C. taylorii mucilage on native benthic communities exposed to other common stressors is strictly required.

In this paper, following a full-factorial design, nutrients were added to simulate eutrophication, macroalgae (both erect and crustose) were removed to simulate barrens produced by grazers and mucilage was manually removed to simulate mucilage-free conditions. We predict that the presence of mucilage would buffer the effect of nutrient enrichment on macroalgal abundance (antagonistic effect). On the contrary, mucilage would reasonably worsen the effects of herbivory, inhibiting macroalgal recovery after massive grazing events (total macroalgal removal) and enhancing the development of permanent barrens (synergistic effect). Therefore, the effect of mucilage was expected to be higher in non-enriched conditions especially when herbivores where present (Burkepile & Hay, 2006; Guarnieri et al., 2014).

Material and Methods

Study site and experimental design

The manipulative field experiment was conducted in Tavolara Punta Coda Cavallo Marine Protected Area (hereafter TPCC MPA, North-East Sardinia, Western Mediterranean), during the summer of 2014. Punta Don Diego Bay (40°52′34.62″N; 9°39′21.19″E), located in a partially protected zone of the MPA (traditional economic activities as fishing and recreation are allowed), was chosen for the experiment as C. taylorii blooms have been recurrently abundant there in the recent years (Caronni et al., 2014) (Fig. 1). The study area is characterized by oligotrophic waters (P: 0.006 ±  0.0003 µM; N: 0.3 ± 0.008 µM) and a well-developed and diversified macroalgal community is present. The main grazers in the area are sea-urchins (Paracentrotus lividus; Lamarck, 1816), whose grazing may cause important changes in the distribution patterns of benthic communities, exerting a paramount role in the transition from macroalgal beds to barrens where only a few coralline algae are present (Boudouresque & Verlaque, 2001; Hereu et al., 2004). Urchin density in the study area is of about 3.5 ind m−2 (P. Panzalis, pers. comm., 2013).

Figure 1 Localization of the study area in Tavolara Punta Coda Cavallo Marine Protected Area.

The differently protected zones of the MPA are also indicated: light grey indicates C zones; middle grey B zones and dark grey A zones.

Two rocky areas of about 10 m2 (20 m apart), comparable in water motion, topography, and inclination of the rocky substrate, were randomly chosen in the bay at 1.5 m of depth (highest C. taylorii cell density depth, (Caronni et al., 2015). In each area 18 plots 20x20 cm in size were prepared and randomly assigned to one treatment.

The experimental design consisted of three factors: three grazing levels crossed with two nutrient enrichment and two mucilage removal treatments, following a factorial design, and three replicates for each combination of treatments were considered (N = 36). To obtain such treatments, plots were differently manipulated at the beginning of July: 1. the mucilaginous aggregates were manually removed from half of the plots (M-), while mucilage was maintained in the other half of them (M+); 2. the substratum was scraped at three levels: total (G100%: all the surface was fully scraped), partial (G50%: the 50% of the surface was fully scraped) and no (G0%) removal of macroalgae (both erect and crustose) was conducted, using an iron brush to simulate the effect of grazers such as P. lividus, responsible for the formation of extended sea urchin barrens (Hereu, 2005); 3. nutrient enrichment was obtained in half of the plots, all in the same area, to avoid nutrient enrichment of control plots (E+ vs E- refereed to enriched and non-enriched) (N = 36). For nutrient enrichment, small-mesh nylon bags (2 mm mesh size) filled with slow-release fertilizer pellets (Osmocote®; 18:9:10, N:P:K) were used, following Bulleri, Russell & Connell (2012) and Guarnieri et al. (2014). The bags were fixed to a brick and positioned on the rocky bottom at the edge of each unit. Overall, 40 g of fertilizer were added in each plot, placing two bags with 20 g of pellet at two opposite sides of the plot. The amount of fertilizer in each bag was decided according to previous studies (Worm & Sommer, 2000). To ensure enriched conditions throughout the experiment, nutrient bags were monthly replaced.

The concentration of nutrients (N and P) in the water was estimated two times from July to September 2014; 10 water samples (125 ml) in each area were randomly taken in July (S1) and 4 weeks later, in August (S2). Samples were taken at approximately 10 cm from the bottom and at least 50 cm apart from nutrient bags. After collection, water samples were shaken, filtered (0.45-µm mesh size filter) and frozen, as suggested by Balata et al. (2010). They were transported to the University of Milan Bicocca, where concentrations of inorganic N and P (ammonia, nitrate, nitrite and phosphate) were estimated using a Spectrophotometer Lambda EZ201 Perkin Elmer (precision=0.3 nm). Chemical analyses of water samples confirmed that for both nutrients differences due to the enrichment were significant and consistent before and 4 weeks after the manipulation (Table 1).

Table 1 Nutrient enrichment effectiveness.

Mean nutrient (inorganic N and P) concentration between nutrient addition and control plots (E+ and E-) on each sampling time.

Time	Treatment	Inorganic N (µM)	Inorganic P (µM)	
		Mean (n = 10)	Stan. dev.	t-test (P)	Mean (n = 10)	Stan. dev.	t-test (P)	
8th week	E+	0.720	0.00039	0.0001	0.170	0.00027	0.0003	
8th week	E-	0.290	0.00035	0.001	0.00013	
36th week	E+	0.780	0.00034	0.0002	0.190	0.00031	0.0001	
36th week	E-	0.320	0.00021	0.010	0.00021	

After an initial sampling, performed at the beginning of July, before the manipulation, in each plot benthic assemblages were sampled two times; 8 weeks later (at the beginning of September 2014, ) and 36 weeks later (in March 2015), to evaluate the short and long-term effects of the three stressors, respectively.

The assemblages in each plot were sampled photographically using a Nikon Coolpix AW130 underwater camera (16 Megapixel, Tokyo, Japan). The camera was held at a consistent distance from the substrate (approximately 50 cm) as suggested by Jonker, Johns & Osborne (2008), paying attention to frame the entire experimental plot. Applying image analysis tools, the percent cover (%) of each macroalgal taxon was assessed, superimposing a grid of twenty-five sub-quadrats onto each image, scoring each sub-quadrat from 0 to 4% and adding the 25 resulting values to obtain the total cover (Dethier et al., 1993).

Statistical analyses

A distance-based permutational multivariate analysis of variance (PERMANOVA) (Anderson, 2001) was performed using the software PERMANOVA (Anderson, 2005), to analyse the response of the macroalgal assemblage to experimental conditions across time. The analyses were based on Bray–Curtis dissimilarities calculated on non-transformed data. Each term in the analysis was tested using 9,999 random permutations. To test for short and long-term effects of treatments and to analyse independent data (Underwood, 1997), two PERMANOVAs were performed on data collected 8 and 36 weeks after the manipulation), respectively. The experimental design consisted of three factors: nutrient enrichment (two levels, fixed), grazing exclusion (three levels, fixed and orthogonal) and mucilage removal (two levels, fixed and orthogonal). Significant terms relevant to the hypotheses were investigated through post hoc pair-wise comparisons and a SIMPER test was also run (Primer v6) to point out the relative contribution of each taxon to the dissimilarities observed among treatments (Clarke, 1993). Finally, a non-metric multidimensional scaling (nMDS) was used for the graphical ordination of data.

To investigate the differences among treatments evidenced by the SIMPER test for the main macroalgal taxa, a three-way ANOVA was also run for each taxon abundance using the software GMAV5 (University of Sydney, Australia). Cochran’s test was run prior to each ANOVA to test for homogeneity of variances and normality was assured by Kolmogorov–Smirnov test. Student–Newman–Keuls (SNK) tests were used for a posteriori comparison of means (Underwood, 1997).

Results

Short-term macroalgal response to disturbance

At the end of the experiment, 8 weeks after the manipulation, eight macroalgal taxa/morphological groups were found: Acetabularia acetabulum (L.) Silva., Dark filamentous algae (DFA), Dasycladus vermicularis (Scopoli) Krasser, Crustose coralline algae (CCA), Dictyota spp. Laurencia spp., Liagora spp., and Padina pavonica (L.) Thivy. The combination of three analyzed stressors affected the short-term recovery of assemblages (PERMANOVA significant E ×G ×M interaction, Table 2). Pair-wise comparisons showed statistically significant differences between enriched and non-enriched plots only when the other two stressors were absent or when their effect was poor (M-G0% and M-G50%). Furthermore, in non-enriched conditions statistically significant differences due to mucilage were recorded, especially in plots where the grazing pressure was high (E-M+G100%; ≠E-M-G100%). On the contrary, the same effect was not observed in enriched plots. Additionally, the effect of mucilage was not detected where the grazing pressure was null or low, in both enriched and non-enriched plots (E+G0% and E-G0%), as clearly depicted in the nMDS graphs (Fig. 2).

Table 2 PERMANOVA results at short-term (8 weeks).

Results of permutational multivariate analyses of variance (PERMANOVA) testing the effect of nutrient enrichment (E), grazing exclusion (G) and mucilage removal (M) on the structure of macroalgal assemblages at short-term (8 weeks). Analyses were based on Bray–Curtis dissimilarities and each test was performed using 9.999 permutations of appropriate units. Significant P-values are given in bold.

Source of variation	df	SS	MS	F	P (perm.)	
Nutrient enrichment (E)	1	1,538.0381	1,538.0381	23.3065	0.0001	
Grazing exclusion (G)	2	3,080.7599	1,540.3799	23.3420	0.0001	
Mucilage removal (M)	1	246.5692	246.5692	3.7364	0.0348	
E×G	2	1,428.6083	714.3041	10.8241	0.0001	
E×M	1	192.8727	192.8727	2.9227	0.0656	
G×M	2	292.5294	146.2647	2.2164	0.0824	
E×G×M	2	286.0645	143.0322	2.1674	0.0472	
Residual	24	1,583.8026	65.9918			
Total	35	8,649.2446				

Figure 2 Experimental design and nMDS.

In the experimental design (A) three grazing levels were considered (G100%, G50% and G0% referred to total, partial and no removal of macroalgae; for nutrient enrichment two levels were considered (E+ vs E- referred to enriched and non-enriched) as well as for mucilage (M+ vs M- referred to plots with and without mucilage). nMDS was performed on short- (8th week, B) and long-term (36th week, C) data. nMDS ordination was not significant on short-term.

Differences in the composition of macroalgal assemblages were found to be related to all three considered stressors and the Simper test evidenced that four taxa remarkably contributed to the observed dissimilarities: dark filamentous algae, crustose coralline algae, Dictyota spp. and Laurencia spp.

Finally, all the ANOVAs performed on these taxa (DFA, CCA, Dictyota spp. and Laurencia spp.) detected differences for the E ×G ×M interaction term (Tables 3 and 4; Fig. 3). Particularly, dark filamentous algae seemed to be more abundant in enriched plots, especially where mucilage and macroalgae had been removed (E+M-G100%), while a higher percent cover of crustose coralline algae was observed where nutrients had been increased especially if the grazing pressure was null, independently from mucilage presence (E+M-G0% and E+M+G0%). Finally, both the erect species (Dictyota spp. and Laurencia spp.) were more abundant in enriched plots, where the grazing pressure was null or low and mucilage had been removed (E+M-G0% and E+M-G50%) (Tables 3 and 4; Fig. 3).

Table 3 ANOVA results.

Results of ANOVAs on the effect of each treatment (nutrient enrichment (E), grazing exclusion (G) and mucilage removal (M) on the percent cover of DFA, CCA, Dictyota spp. and Laurencia spp. at short-term (8 weeks). Significant P-values are given in bold.

Source of variation		DFA	CCA	Dictyotales	Laurencia spp.	
	df	F	P	F	P	F	P	F	P	
Nutrient enrichment (E)	1	5.68	0.0254	69.84	0.0000	0.52	0.0471	67.83	0.0000	
Grazing exclusion (G)	2	2.24	0.0388	32.29	0.0000	37.10	0.0000	48.23	0.0000	
Mucilage removal (M)	1	7.36	0.0122	1.92	0.0383	0.12	0.0252	0.25	0.0213	
E×G	2	9.06	0.0012	8.52	0.0016	9.60	0.0009	22.80	0.0000	
E×M	1	9.06	0.0012	0.32	0.5792	0.01	0.9102	8.75	0.0069	
G×M	2	3.47	0.0473	3.62	0.0424	0.45	0.6451	0.54	0.5918	
E×G×M	2	1.88	0.0487	2.71	0.0469	0.12	0.0490	1.37	0.0373	
Residual	24									
Total	35									
Cochran’s test (C)		0.3556 (NS)	0.3542 (NS)	0.3453 (NS)	0.3996 (NS)	

Table 4 SNK results.

Results of SNK test on ExGxM interaction for the 4 main taxa for: nutrient enrichment (E+ vs E-), grazing exclusion (G100%, G50% and G0%) and mucilage removal (M+ vs M-).

DFA
Grazing exclusion
M+
E+ G100%>G50%=G0%
E- G100%=G50%=G0%
M-
E+ G100%>G50%>G0%
E- G100%=G50%>G0%	Mucilage removal
G100%
E+ M+<M-
E- M+=M-
G50%
E+ M+=M-
E- M+=M-
G0%
E+ M+=M-
E- M+=M-	Nutrient enrichment
G100%
M+ E+=E-
M- E+>E-
G50%
M+ E+=E-
M- E+>E-
G0%
M+ E+=E-
M- E+=E-	
CCA
Grazing exclusion
M+
E+ G100%<G50%<G0%
E- G100%=G50%=G0%
M-
E+ G100%<G50%<G0%
E- G100%=G50%=G0%	Mucilage removal
G100%
E+ M+=M-
E- M+=M-
G50%
E+ M+=M-
E- M+=M-
G0%
E+ M+<M-
E- M+=M-	Nutrient enrichment
G100%
M+ E+=E-
M- E+=E-
G50%
M+ E+=E-
M- E+=E-
G0%
M+ E+>E-
M- E+>E-	
Dictyotales
Grazing exclusion
M+
E+ G100%=G50%=G0%
E- G100%=G50%=G0%
M-
E+ G100%<G50%=G0%
E- G100%=G50%=G0%	Mucilage removal
G100%
E+ M+=M-
E- M+=M-
G50%
E+ M+<M-
E- M+=M-
G0%
E+ M+<M-
E- M+=M-	Nutrient enrichment
G100%
M+ E+=E-
M- E+=E-
G50%
M+ E+=E-
M- E+>E-
G0%
M+ E+=E-
M- E+>E-	
Laurenciaspp.
Grazing exclusion
M+
E+ G100%=G50%=G0%
E- G100%=G50%=G0%
M-
E+ G100%<G50%=G0%
E- G100%=G50%=G0%	Mucilage removal
G100%
E+ M+=M-
E- M+=M-
G50%
E+ M+<M-
E- M+=M-
G0%
E+ M+<M-
E- M+=M-	Nutrient enrichment
G100%
M+ E+=E-
M- E+=E-
G50%
M+ E+=E-
M- E+>E-
G0%
M+ E+=E-
M- E+>E-	

Long-term macroalgal response to disturbance

Thirty-six weeks after the manipulation only six macroalgal taxa/groups were overall found and the effect of treatments (nutrient enrichment, grazing exclusion and mucilage removal) was neither highlighted on the most abundant algae (DFA, CCA, Dictyota spp., Laurencia spp. and P. pavonica) nor on the whole structure of the macroalgal assemblages, as graphically evidenced also by the nMDS ordination (Fig. 2).

Discussion

An interactive effect of the three stressors on the short-term recovery (8 weeks after the manipulation) of the considered macroalgal assemblages was highlighted in the study (Figs. 2 and 3; Tables 3 and 4). Significant differences in the composition and abundance of macroalgae were observed depending on enrichment, but only when mucilage and grazing were absent (Figs. 2 and 3; Tables 3 and 4). These results are in accordance with those of other studies investigating the role of nutrient enrichment in determining macroalgal abundance (e.g., McGlathery, 2001; Teichberg et al., 2008; Sotka & Hay, 2009) and confirm that, when nutrient enrichment is the only stressor, it leads to a remarkable increase of total macroalgal biomass. More in detail, a positive response to nutrient enrichment of the different macroalgal taxa present in the study area was observed, with an enhancing effect on both turf-forming and erect macroalgae. Our evidence supports the results obtained by Bulleri, Russell & Connell (2012) who found a positive effect of enrichment on all macroalgae, contrarily to Pedersen & Borum (1996), who observed an enhancement of turf-forming algae, in particular of DFA.

Figure 3 Main taxa percent cover.

Percent cover (mean % ± SE, n = 20) of the 4 main taxa (dark filamentous algae (DFA), crustose coralline algae (CCA), Dictyota spp. and Laurencia spp.) for each combination of treatments: nutrient enrichment (E+ and E-); grazing (100%, 50% and 0% of macroalgal removal); mucilage (M+ and M-)).

Instead, no relevant differences were observed in the composition of macroalgal assemblages exposed to nutrient enrichment when one of the other two stressors was present (Figs. 2 and 3; Tables 3 and 4). In particular, a buffering effect of mucilage on nutrient enrichment was recorded. In plots where mucilage was present, no significant differences in macroalgal assemblages’ abundance and composition were noticed between enriched and non-enriched treatments (Figs. 2 and 3; Tables 3 and 4). Results can be explained by considering that microalgae take up nutrients faster than macroalgae and consequently mucilage might be responsible for the sequestration of large amounts of nutrients from the water column, elements then used by microalgae embedded in aggregates to survive and proliferate (Reynolds, 2007). As a matter of fact, aggregates are biota-rich environments where the concentration of nutrients can be dramatically higher than in the surrounding seawater (Del Negro et al., 2005). For this reason, it is plausible that, in plots with mucilage, only a small amount of total nutrients released in the water was available for macroalgae as a conspicuous portion of them was sequestrated and used by mucilage. Furthermore, even if Huang & Boney (1983) observed that, in laboratory conditions, the growth of some species of green and brown algae was enhanced by diatoms mucilage, mucilaginous aggregates are generally known to overgrow macroalgae causing their mechanical suffocation and rapid biomass depletion (Misic, Schiaparelli & Covazzi Harriague, 2011). Moreover, Müller et al. (1998) assumed that all benthic organisms are seriously damaged by mucilage aggregates, even if their cover is thin because they usually release toxins directly affecting cell metabolism. Therefore, when mucilage and nutrient enrichment acted simultaneously, the expected increase in macroalgal abundance due to nutrient enrichment could have been counterbalanced by the presence of mucilage on the substratum.

The positive effect of nutrient enrichment did not seem buffered by mucilage; only considering crustose coralline algae, their abundance was equal in all enriched plots, independently from the presence of mucilage (Figs. 2 and 3; Tables 3 and 4). Schiaparelli et al. (2007) and Figueiredo & Steneck (2000) suggested that coralline algae could be seriously damaged by mucilage (especially when its presence on the substratum lasted for a long time), but our results mostly agree with Bulleri (2006) who evidenced their ability to survive for long periods if overgrown by other species.

As for mucilage, also the effect of grazing seemed to buffer that of nutrient enrichment (Figs. 2 and 3; Tables 3 and 4). About this, Guarnieri et al. (2014) observed a relatively constant macroalgal cover, also in nutrient enriched conditions, when a high grazing pressure acted. Our results highlight that the presence of herbivores can strongly affect the proliferation of macroalgae, lowering the positive effect of enrichment. Therefore, even if the results of several previous experiments suggested that both increased nutrient loading and reduced grazer densities favour an intense macroalgal growth (e.g., Geertz-Hansen et al., 1993; Hauxwell et al., 1998; Lotze & Worm, 2000), the reduction of herbivory often seems to be the main factor triggering the restoration process of macroalgal assemblages after disturbance (Scheffer et al., 2001), and the role of nutrient enrichment appears to be only secondary. Nevertheless, herbivory does not represent the exclusive process in structuring macroalgal assemblages, as our results highlight its interactive effect with nutrient availability in mediating the outcomes of grazing pressure, similarly to Burkepile & Hay (2006). In this interaction, the mucilage plays a relevant role that should not be neglected. In fact, where the grazing pressure was null or low the presence of mucilage did not influence the abundance of the main taxa (Figs. 2 and 3; Tables 3 and 4) except for where enrichment was done. This suggests a primary role of herbivores and nutrients in regulating the abundance of macroalgae (Lawrence, 1975; Underwood, 1980; Scheibling, 1986; Geertz-Hansen et al., 1993). However, mucilage effect is corroborated by the effect of enrichment on Laurencia spp. and Dictyotales, at no grazing pressure, only where the mucilage was removed. This would support the hypothesis that the effect of enhancement produced by nutrient enrichment can only be conspicuous where the negative effect of mucilage was nullified.

Therefore, the effect of mucilage seems to be significantly detrimental only where communities are stressed by high densities of herbivores and nutrients are not so abundant to remarkably increase macroalgal biomass. In such conditions, indeed, damages caused by both mucilage suffocation and mechanical detachment of macroalgae (especially of erect and frondose species) (Schiaparelli et al., 2007; Lugliè et al., 2008) are not worsened by grazing pressure and balanced by nutrient enrichment. Also, the lower abundance of erect species such as Dictyotales and Laurencia spp. confirms that the erect species are more damaged by mucilage than the turf-forming ones. Furthermore, turf-forming species appeared to be more abundant also where macroalgae were removed because, in such conditions, a wider free space was available. Furthermore, in this last case, lower competition favoured the development of opportunistic species such as turf-forming macroalgae (Bulleri, Russell & Connell, 2012).

Finally, the lack of effects of the three considered stressors observed after about eight months of study deserves a deeper insight. Whether the change through time of effects was biased by a seasonal effect (because of the variable phenology of algal taxa) it will remain unknown because unfortunately no data are available at present. Therefore, very hard disentangling mechanisms (seasonal effect vs lack of complete recover) are responsible for the lack of persistent differences through time. However, even if the variability of effects through the year could occur depending on the composing species seasonal performance, there are reasons to hypothesize that the macroalgal assemblage would cope with the considered stressors assisted by cyclical responses.

Conclusions

The short-term effect (8 weeks) of the considered multiple stressors on the macroalgal community suggest the mucilage and grazing to be antagonistic to nutrient enrichment, while mucilage presence and herbivore pressure seem to act synergistically. Particularly, dark filamentous algae were more abundant in enriched plots, especially where mucilage and macroalgae had been removed; conversely, a higher cover of crustose coralline algae was observed where nutrients had been increased and no grazing pressure acted. Furthermore, the abundance of Dictyota spp. and Laurencia spp. was significantly higher in enriched mucilage-free plots where the grazing pressure was null or low.

However, effects of treatments on the overall assemblage of the macroalgal community were not persistent on the longer-term (36 weeks after the manipulation) and this indicates the capacity of a shallow-water macroalgal community to quickly recover from these simultaneous impacts. Therefore, even if these results substantially confirm the predictions on the interactive stressors (mucilage would have buffered the effect of nutrient enrichment, worsened the one of herbivory and that a greater effect of mucilage would have been observed in non-enriched conditions especially when herbivores were present), the rapid fading of their effects in a few months makes it not strictly necessary or urgent to lay out guidelines for managers.

At the same time, the results of this study can also help to forecast areas where nutrient pollution will have more detrimental effects on macroalgal assemblages, basing on the grazers density and the frequency of mucilaginous blooms. Furthermore, some interesting ecological lessons can also be gained, as relevant information has been provided about the resilience of macroalgal communities to mucilage effects at non-pristine conditions, which concerns the nutrient water status and the herbivores control by predators. In such conditions indeed the effects of disturbances, even when they are due by multiple stressors acting together, may be more efficiently buffered within the system (consistently to Agardy (1994) and to Jentoft, Van Son & Bjørkan (2007)), highlighting that stability of the state would be provided by other feed-back mechanisms.

Supplemental Information

Dataset S1 Complete dataset of all raw data

The file contains the complete detailed raw data regarding the percent cover of different macroalgae on the substratum in the 3 sampling times (T0-T2).

Click here for additional data file.

We thank the two anonymous referees and the editor Craig Nelson for providing useful suggestions to improve the ms. Furthermore, we thank Luisa Polastro, Sidney Freedman and Nadia Carlier for the English revision, Luigi Piazzi for the fundamental help in data analysis and Matteo Grechi and Gianluca Cavagna for the precious help during image analysis.

Additional Information and Declarations

Competing Interests

Author Contributions

Data Availability

The authors declare there are no competing interests.

Sarah Caronni conceived and designed the experiments, performed the experiments, analyzed the data, prepared figures and/or tables, authored or reviewed drafts of the paper, approved the final draft.

Chiara Calabretti, Maria Anna Delaria and Giovanni Macri performed the experiments, approved the final draft.

Sandra Citterio contributed reagents/materials/analysis tools, authored or reviewed drafts of the paper, approved the final draft.

Rodolfo Gentili prepared figures and/or tables, approved the final draft.

Chiara Montagnani authored or reviewed drafts of the paper, approved the final draft, enghish editing post suggestions of the professional language editing service of our university.

Augusto Navone and Pieraugusto Panzalis contributed reagents/materials/analysis tools, approved the final draft.

Giulia Piazza contributed reagents/materials/analysis tools, prepared figures and/or tables, revision.

Giulia Ceccherelli conceived and designed the experiments, performed the experiments, analyzed the data, contributed reagents/materials/analysis tools, authored or reviewed drafts of the paper, approved the final draft.

The following information was supplied regarding data availability:

The raw data are available in a Supplemental File.

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
