# Peer review of "The interactive effect of herbivory, nutrient enrichment and mucilage on shallow rocky macroalgal communities"

_PeerJ, doi:10.7717/peerj.6908_

## Round 0.1 · original submission · Major Revisions

Your article has now been reviewed thoroughly by two scientists, both of whom bring direct experience working on intertidal macroalgal community ecology. Their input is significant, and it is my view that your manuscript has potential but is unacceptable in its current form. Addressing the central criticisms, particularly with respect to statistical analysis and the various issues raised regarding graphs and visualization of the data, will certainly strengthen the manuscript, hopefully to the level necessary to make it publishable in PeerJ. However, I encourage you to focus effort on selecting and justifying appropriate statistical tests to answer the questions posed: the reviewers have both recommended particular approaches, with Reviewer 2 taking great care to identify recommended methods to use on your data to make the analyses robust. It is possible that these methods, or analogous methods that you select and justify, will alter the conclusions of the manuscript and may change how you choose to display the data; this should not dissuade you, and you are welcome to contact me or the journal if you wish to submit a new version rather than a revision. I also recommend that any revision or resubmission address the english language issues raised by the reviewers and also conduct a thorough revision with input from a native speaker, as PeerJ does not offer copy editing services.

Reviewer 1 ·

Basic reporting

Your introduction needs more detail. I suggest a more in-depth look and representation of the literature on the specific effects of the factors you are testing in this study, especially mucilage. Some citations span ecosystems other than shallow rocky shores, which was good, but did not include some of the more relevant papers such as Smith et al. 2001, 2010 etc. Certain concepts were not explained in the introduction, including: the relationship between mucilage abundance and “blooms”.

I thank you for providing the raw data, however your supplemental files need more descriptive metadata identifiers to be useful to future readers, specifically an explanation of the Time, Unit, and Treatment codes is not included.

There is an unlabeled table on page 32. Is this a supplemental table? It needs a caption.

There seem to be no standard error bars on the inorganic P bars on Figure 1. Please add them, if they are not there.

The caption for Figure 2 indicates that the MDS plot should have black or white symbols with shapes around them but the actual figure only has labels.

Figure 3 does not have a legend for the bars’ color patterns, which should be associated with mucilage (M+ and M-).

In line 47 the (Lewis & Bryan 1941) citation is not necessary, as this is the abstract.

In lines 52-57, the responses are unclear. This section should explicitly discuss the results of the study.

Include a reference to the appropriate table or figure when discussing results (in the discussion section).

Line 106, please explain what you mean by “interact detrimentally”.

Line 112, the correct citation is (Lewis & Bryan 1941) within the text. Please correct the Silva citation on line 188 and the Thivy citation on line 190.

Line 131, please define C zone.

The sentence that spans line 157-159 belongs in the results section.

Please add the year for Huang & Boney study in line 250.

Lines 302-305 should be re-written for clarity.

According to the author guidelines, the acknowledgement section should not be used to acknowledge funders. This information will appear in a separate Funding Statement on the published paper.

Experimental design

Please include a map of the MPA to make visualization of the study area easier. In lines 134-137, you describe the experimental design. It would be helpful to include a panel in the map figure that shows a conceptual representation of the experimental design as well.

In line 157, the authors wrote that the “the concentrations of inorganic N and P (ammonia, nitrate, nitrite and phosphate) were estimated”. Please describe the chemical analyses, including the instrument(s) used for these analyses. The precision of the instrument’s measurements should also be included.

Were the plots surrounded by cages in order to exclude real herbivory, considering one of your treatments was artificial grazing? How did you control for herbivory after the artificial grazing treatment was imposed?

Validity of the findings

There is no evidence (in the form of a figure or table) for the results presented in lines 192-199. There is an unlabeled table on page 32, which may contain this information, but without a caption, it is impossible to tell what the information in the table is conveying.

It would be useful to have a table with the SIMPER results to show the percentages of the differences in assemblage associated with the dark filamentous algae, encrusting coralline algae, and macroalgae (lines 200-204).

The discussion in lines 214-217 on the long-term macroalgal response to disturbance needs more explanation. It is unclear that Table 2 has to do with the second sampling effort at T2 (line 217).

The discussion, with some grammatical and English language clean-up, does a good job of explaining your results in the context of some of the literature. If you could expand on the discussion of the importance or implications of the findings as it pertains to MPAs, the field of study, or management in the conclusion section, this would strengthen this section and make it relevant to a broader audience.

It is important to include a discussion of the limitations of the study in the discussion section (eg. small sample size, seasonal variance, etc.).

In the text of the results please discuss the quantitative enrichment of N and P in the nutrient treatments. N was enriched about 2 times the concentration of the water around the non-enriched plots, while P was enriched much more. Discuss how this may have affected your results, if at all.

Additional comments

I suggest the authors get editing help from someone with full professional proficiency in English.

A more in-depth discussion of the results is required with accompanying evidence in the form of relevant figures and tables.

I enjoyed reading your article. I learned a lot about mucilage and its interactive effects with nutrients and herbivory ☺.

Reviewer 2 ·

Basic reporting

The manuscript submitted by Caronni et al. seeks to disentangle the interactive effects of three stressors on the benthic algal composition of rocky reefs in the eastern Mediterranean. This study was motivated by the increasing occurrence of large blooms of a mucilage producing microalga that can have negative impacts on nearshore marine communities. The study intended to assess the relative impact of herbivory, nutrient enrichment, and the presence of mucilage on the composition of benthic algae. While the results of this work will likely be valuable to local management agencies seeking ways to reduce the spread and impact of this species, there are multiple issues with this manuscript that must be addressed before it can be considered for publication. Foremost, I do not believe the statistical analyses are adequate to support the conclusions of the study. The figures are also poorly developed and further hinder the ability of the reader to assess the experimental data. There are also multiple locations where the literature is incorrectly cited and there is a lack of detail in the introduction and discussion that make understanding the relative importance of the study challenging. Finally, while the use of professional English is generally acceptable, there are a number of places where sentences are difficult to follow and an overabundance of qualifying words. Please see the general comments sections for specific examples of these issues.

Experimental design

The overall aims of the study are clearly defined but I think the authors could do a better job at identifying the knowledge gap filled by their study through a more comprehensive discussion of their study system. Also, the description of the experimental design has a few issues that must be addressed. The most important is more clearly describing the level of replication and spatial arrangement of the experimental plots.

Lines 137 – 150: Understanding the level of replication and the spatial arrangement of treatments in this experiment would be easier if the authors provided a supplemental graphic or table.

Lines 139 – 141: Helping contextualize the study system for the reader will go a long way to improving the accessibility of this study to people unfamiliar with the study system. For example, what are the dominant grazers here? And how do they impact the algal communities? It seems like using a wire brush to completely remove all algae is an extreme grazing treatment but perhaps it is not for this location.

Lines 141-149: The nutrient enrichment treatments in this study are not that high with respect to
“eutrophication stress.” It would be helpful if the authors provided some background about the oceanography of this region and some existing values of nutrient impacted sites nearby. This would help contextualize the ecological relevance of the nutrient enrichment treatments. In an algal dominated system nitrate levels of 1-1.5 µmol can be rapidly utilized. The authors should also discuss if their in situ measurements reflect a residual nutrient pool that represents all of the nutrients not used by the algae or if they believe that this enrichment reflects their constant dosing concentration.

Validity of the findings

The validity of the overall findings of this study are difficult to assess. A large part of this may be due to the lack of detail reported in the methods and the author’s interpretation of the statistical results. Additionally, the figures in this manuscript are poorly developed and inadequately labeled, which makes it hard to visualize and interpret the results. Below I provide specific comments where these aspects of the study may be improved.

Lines 140 – 141: Grazing pressure is an important variable in this study but I think that it is confounded in the statistical analysis of benthic community composition. The complete removal of erect macroalgae will undoubtedly alter the benthic composition of the “high grazing” treatment after only 3 months, when the benthic data for “short term” effects is presented. Consequently, the statistical assessment of your grazing effect needs to be handled differently.

To broadly compare species composition across grazing treatments is incorrect. One way you could get around this is by comparing the relative change in species cover between your time points. For example, between July (right after plots were cleared) and September (3 months recovery) and then again from September to March to assess longer term impacts on community composition.

Lines 148-150: One of the more surprising results of this study was the lack of persistent effects of these treatments over longer time scales. The authors did not discuss this important result in much detail and based on the reporting of the methods it is hard to understand whether this result is due to an experimental artifact or natural processes in the region.

Notably, it is not confirmed that the treatments (mainly nutrient enrichment) are maintained from September to March. This is a critical piece of information for the reader to determine if the effects of nutrient enrichment had no effect at longer time scales. As written, I am led to believe that the data from March assess the relative impact of a short-term manipulation rather than a persistent, press disturbance of nutrients through time.

Lines 162-167: The only response variable considered in this study is the proportional composition of benthic algal communities in each plot. As such, a more rigorous and quantitative approach to assessing these changes is necessary. There is no detail provided about how these photographs were taken (e.g., what distance from the bottom?, did the experimental plot fill the full frame of the photograph?). This information is important because it will determine how comparable photographs are between plots. Additionally, the qualitative assignment of visual percentages based on 25 mini-quadrats is likely to introduce a lot of noise into the data. This is important because it might mask important differences between treatments that might otherwise be missed. I would suggest the authors re-analyze the photographs (there are very few) using a more quantitative software such as Coral Net (https://coralnet.ucsd.edu/) or CPCE (https://cnso.nova.edu/cpce/index.html).

Statistical analyses and figures

As mentioned above, I think that the response variable of this study should be re-considered due to the confound nature of the grazing treatment on the percent cover metric. Assessing the relative change from To (July) to T1 and T2 is one way around this problem.

The statistical tests used to assess the effects of this study make interpretation of the results challenging for both the reader and the authors. Foremost, a three-way interaction in the PERMANOVA means that the direct effects of each treatment cannot be considered independently as is often done in the discussion. Following a PERMANOVA with a SIMPER analysis, independent ANOVAs, and the displaying the data in an nMDS is also not the best approach. I would recommend using CAP analysis to visually represent each of the treatments and simultaneously examine the relative contribution of the dominant benthic taxa to these differences. See Anderson and Willis, 2003. Canonical analysis of principal coordinate: a useful method of constrained ordination for ecology. Ecology 84: 511-525.

Additionally, to more robustly investigate the short vs. long-term impacts of the multiple treatments the authors might consider calculating successional trajectories in multivariate space. See Smith et al. 2010. The effects of top-down versus bottom-up control on benthic coral reef community structure. Oecologia DOI 10.1007/s00442-009-1546-z.
Finally, the figure captions and figures of this manuscript need to be improved. In figure 3 there is no explanation of what the different shaded bars represent for each of the grazing treatments. Do these correspond to the mucilage treatment or to T0 and T1? Figure 2, as presented is impossible to interpret. The authors should remake this figure using shapes and colors to help the reader visualize the results of the study. Additionally, a similar figure (pending updated analysis) should also be shown for the sampling in March (t2), which will help show how these plots changed through time.

By improving the analysis and refining their statistical approaches, the authors will be able to more clearly dictate their findings and will ultimately present a more compelling narrative. Multiple stressor experiments are inherently difficult and I commend the authors for trying to assess such an important topic. However, care must be taken in the interpretation of the statistical results of these complex experiments. I hope these suggestions will help the authors improve future versions of this manuscript.

Additional comments

Overall, I think this study will provide important information about the benthic communities and the respective impacts of several natural stressors in the eastern Mediterranean. In order to improve the accessibility of this study to readers unfamiliar with the area, the authors should work to clarify their key results and place them in a broader ecological context. Additionally, make sure to use the literature accurately and to cite some more recent papers that have considered some of the topics addressed in your study. Below are a few line by line comments that I hope will aid in the revision of the manuscript.


Lines 45-46: the country/ocean of the experimental area should be provided.

Line 54: these abbreviations must be defined if they are going to be included in the abstract

Lines 62 – 70: Watch out for run-on sentences. Also, there has been considerable work on the impacts and interpretation of multiple stressor experiments in recent years. I think the introduction as a whole would benefit from the inclusion of some more recent literature. But with respect to the last sentence of this paragraph, I would suggest the authors consider recent reviews on multiple stressor impacts, such as Kroeker et al. Embracing interactions in ocean acidification research: confronting multiple stressor scenarios and context dependence. Biology letters 13.

Lines 72-73: There are multiple instances throughout the manuscript where qualifying words should be removed. For example, here the words “merely” and “mere” and unnecessary.

Lines 77-84: The literature cited in this manuscript is often misused. Here the Koch and Peckol Rivers papers are not appropriate references for the opening sentence. The authors should see studies that more quantitatively assess the foundational role of macroalgae in ecosystem functioning. For example, Graham 2004 Effects of local deforestation on the diversity and structure of southern California giant kelp forest food webs. Ecosystems 7: 341-357.

Line 94: The term mucilage is central to this manuscript and I think it would be beneficial for the authors to describe this characteristic of the dominant mucilage forming taxa at their study site in greater detail.

Lines 95-98: These two sentences appear to contradict each other.

Line 100: Is it the mucilage or the microalgae that overgrows other taxa?

Line 131: What is a c-zone?

Lines 155-159: What methods or instruments were used to quantify nutrient concentrations? The statistical results of nutrient enrichment should be moved to the results section.

Lines 187-190: The taxonomic categories of algae should be moved to the methods. Also, a brief explanation of what genera are included in the broad order of “Dictyotales” should be included.

Line 235: The Fujita paper is about nutrient uptake kinetics in Ulva and Gracillaria. It does not assess anything about relative enhancements of turf-forming algae.

Lines 240-253: The point of these sentences, I believe, is to suggest that perhaps the mucilage forming microalgae take up nutrients faster than macroalgae and therefore buffer the effect of nutrient enrichment. If this is accurate, the authors should streamline and clarify this paragraph as the overall message can be lost on the reader.

Line 254: The paper by Muller is about microalgae growing on snow and their relative photosynthetic pigment concentrations. It is not a study that has anything to do with benthic organisms or mucilage.

---

## Round 0.2 · Major Revisions

Your revision has been received and I have reviewed it, along with the rebuttal letter. Before I return the manuscript to the reviewers I ask that you revise your rebuttal so that EVERY response points to specific pages, lines, figures, etc. where changes have been made to address the reviewer's comment. Such details can drastically ease the burden on the reviewer and improve the likelihood that they will understand and approve the changes you have made.

---

## Round 0.3 · Major Revisions

Both reviewers that reviewed the original manuscript have agreed to re-review this revision. The authors and I both appreciate this dedication on the part of the reviewers, as their reviews have been extensive and very useful.

The revised version has been re-reviewed and both reviewers have indicated significant improvement. However, they both had significant additional questions about the revised manuscript that need to be addressed before final acceptance. I expect that a revision addressing their comments be submitted with equally thorough attention to the points raised as before. Both reviewers were complimentary of the revision effort and the rebuttal, so this should be straightforward.

These are not trivial details, and in particular, both reviewers expressed frustration with the manuscript - it is mandatory that the authors get this edited by a native English speaker before submitting a revised version because it is still very difficult to understand some parts. I cannot stress this enough - please find a native English speaker to clarify the sentence structure.

Also, the CAP analysis is problematic (Figure 4). First, separate analyses should be done on T1 and T2 data because the variance is hugely different and currently T2 is not interpretable on these graphs. Second, if possible this should just be an NMDS. Use taxa correlations with the axes to identify those taxa that drive the NMDS (maybe the top 2-3 for each axis). The current biplot approach (and the CAP method in general) is not appropriate for these data because the taxa simply aren't strongly covariant with the axes.

I recommend that a single NMDS be made for T1 and another for T2, coding the sample points by treatment. I recommend shading for the three G treatments (white, grey, black) and then shapes for the other treatments (open and closed symbols). Then I recommend the authors circle groups of replicates from each treatment that cluster and fit the patterns observed - a figure like this should illustrate the PERMANOVA results visually.

Reviewer 1 ·

Basic reporting

Basic reporting

1.Your introduction needs more detail. I suggest a more in-depth look and representation of the literature on the specific effects of the factors you are testing in this study, especially mucilage. Some citations span ecosystems other than shallow rocky shores, which was good, but did not include some of the more relevant papers such as Smith et al. 2001, 2010 etc. Certain concepts were not explained in the introduction, including: the relationship between mucilage abundance and “blooms”.
DONE: We modified the introduction, deepening on the three factors tested in the study and explaining in detail all concepts. In particular, we added a paragraph on mucilage, focusing on the relationship between mucilage and microalgal blooms (L. 117-133) ad we added the citations of the suggested papers (L. 104, 109, 111).

The introduction is much improved including the citations suggested and with good discussion of the effects of nutrient addition, herbivory, and mucilage on macroalgal communities.

2. I thank you for providing the raw data, however your supplemental files need more descriptive metadata identifiers to be useful to future readers, specifically an explanation of the Time, Unit, and Treatment codes is not included.
DONE: The Excel file containing raw data was reviewed and descriptive metadata regarding Time and Treatment codes were added.

Thank you for addressing this comment.


3. There is an unlabeled table on page 32. Is this a supplemental table? It needs a caption.
We checked the PDF and no unlabeled tables are present on page 32. There is a table that seems not to have the capture on page 29 but actually it is the second part of Table 2 as it reports the results of the SNK test so it is described in the second part of the label regarding Table 2.

I have looked over table 2 in the manuscript and these tables are overlaid and hard to read still but the caption is now helpful in understanding the results presented.

4. There seem to be no standard error bars on the inorganic P bars on Figure 1. Please add them, if they are not there.
We checked bars for Figure 1 (now Fig. 3) and we thank the reviewer because, also if they were present, they were not visible. We changed the color to make them more visible.

I can see the standard error bars now and Figure 3 is more understandable. I have read the responses to the second reviewers comments on nutrient enrichment and that gave me some context on the reasons for the amount of enrichment for both N and P.

5. The caption for Figure 2 indicates that the MDS plot should have black or white symbols with shapes around them but the actual figure only has labels.
DONE: We removed Figure 2 replacing it with Fig. 4, for which a detailed caption is provided.

Great, thanks.

5. Figure 3 does not have a legend for the bars’ color patterns, which should be associated with mucilage (M+ and M-).
DONE: We changed Fig. 3 (now Fig. 5) removing color patterns, using solid colors and adding letters to increase the clarity of the figure.

Figure 5 is much easier to interpret and is well labelled.

6. In line 47 the (Lewis & Bryan 1941) citation is not necessary, as this is the abstract.
DONE: We deleted the citation (L. 46).

Thank you for addressing this comment.

7. In lines 52-57, the responses are unclear. This section should explicitly discuss the results of the study.
DONE: We modified the text to clearly present the main results of the study (L. 51-60).

Thank you for addressing this comment.

8. Include a reference to the appropriate table or figure when discussing results (in the discussion section).
DONE: We added references to tables and figures in the discussion section (L. 282, 285, 299, 302, 323, 328, 342).

Thank you for addressing this point. I think the discussion has improved significantly with added discussions on limitations and broader impacts. It would be nice to explicitly address what your hypotheses were originally and how your results compared to what you originally had hypothesized.

9. Line 106, please explain what you mean by “interact detrimentally”.
DONE: We changed the sentence to better explain the concept (L. 136).

Thank you for addressing this comment.


10. Line 112, the correct citation is (Lewis & Bryan 1941) within the text. Please correct the Silva citation on line 188 and the Thivy citation on line 190.
We are sorry but we don’t agree with the referee. Citations for plants and algae differ from that used for animals. We found these citations on Algaebase (www.algaebase.org) and in a lot of scientific papers (for microalgae please see for example: Monti et al., 2007; Guerrini et al., 2010 and Granéli et al., 2011 while for macroalgae please see for example Menzel, 1981, Serikawa and Mandoli, 1999; Ktari e Guyot, 1999; Campanella et al., 2001; Conti e Cecchetti, 2003):

I am fine with this citation method, if the Editor agrees.

11.Line 131, please define C zone.
DONE: We specified in the text what a C zone is (L. 175-176).

I see you have added a map of the experiment’s location. Please label zone C in this Figure. I have other suggestions for Figure 1 and 2, which can be found in comment number 15.

12.The sentence that spans line 157-159 belongs in the results section.
We preferred to include the sentence in the M&M section as it doesn’t report the results of the experiment and it only describes control data on nutrient enrichment, useful to prove the effectiveness of the applied treatment. Also in other papers this type of results are included in the M&M section. Please see for example Guarnieri et al., 2014.

I understand your rationale and I agree with your choice.

13.Please add the year for Huang & Boney study in line 250.
DONE: We added the year and we also checked the reference section adding the reference regarding this paper that was not present in the previous version of the paper (L. 311).

Thank you for addressing this comment.

14.Lines 302-305 should be re-written for clarity.
PARTIALLY DONE: We didn’t rewrite the sentence but we add some information to clarify it (L. 362-366).

Thank you for addressing this comment.

According to the author guidelines, the acknowledgement section should not be used to acknowledge funders. This information will appear in a separate Funding Statement on the published paper.
DONE: We deleted the sentence regarding funding (L. 393-395).

Thank you for addressing this comment.

Experimental design

15.Please include a map of the MPA to make visualization of the study area easier. In lines 134-137, you describe the experimental design. It would be helpful to include a panel in the map figure that shows a conceptual representation of the experimental design as well.
DONE: We added a map on the MPA with a focus on the study area (Fig. 1) and we added a new figure with a graphical representation of the experimental design (Fig. 2).

Thank you for adding a map (Fig. 1) of the MPA and the study location. I have a few suggestions for this Figure. (1) The enlarged map of the site on the top right corner needs to be zoomed into the bay where you did your manipulative experiment. As it is now, it is hard to see what is in the little white box, which I assume is your actual study site. (2) Please show zone C on the map. (3) Please indicate what the shaded areas represent. A legend would be helpful here.
I appreciate that you included a representation of your experimental design. My one comment on this would be to complete the figure by drawing out all the branches of your factorial design. It would also be helpful to have the N=36 on the actual figure since it’s not said explicitly in the text either.

Were the two rocky areas chosen because 1 area was not enough to house all plots? I think this must be the case considering you randomly assigned treatments to all plots. In your analyses did you consider “area” as a random effect? The one parameter that wasn’t measured in your experiment that could potentially make a difference in terms of mucilage and its physical effects on macroalgal community structure is wave action or water motion. Were these two areas comparable in water motion, topography, and angling of the rocky substrate?

16.In line 157, the authors wrote that the “the concentrations of inorganic N and P (ammonia, nitrate, nitrite and phosphate) were estimated”. Please describe the chemical analyses, including the instrument(s) used for these analyses. The precision of the instrument’s measurements should also be included.
DONE: We insert the requested information in the text (L. 206-207).

Thank you for addressing this comment.

17.Were the plots surrounded by cages in order to exclude real herbivory, considering one of your treatments was artificial grazing? How did you control for herbivory after the artificial grazing treatment was imposed?
No, we didn’t surround plots with cages, because we didn’t consider real herbivory as a problem. The two study areas, indeed, were in the some site and no different conditions regarding herbivores density were present, so no differences in the effect of real grazing potentially affecting results were supposed to exist among units.

Thank you for clarifying herbivore density at the two areas. Is herbivory generally low at these sites? Could you add a sentence about palatability of your species? Even if the herbivore density is comparable at the two sites, some herbivores will most likely have preferences in certain macroalgal species, which could be a confounding factor in your study.

Validity of the findings

1. There is no evidence (in the form of a figure or table) for the results presented in lines 192-199. There is an unlabeled table on page 32, which may contain this information, but without a caption, it is impossible to tell what the information in the table is conveying.
The table that appears to be unlabelled is actually part of Table 2 (as it is indicated in the caption of Table 2) and reports the results of the SNK test for the most abundant species. As pair-wise comparisons were a lot and some of them were not significant, we decided to focus only on significant ones, describing them in detail in the text. Furthermore, the same information provided by comparisons is obtainable observing the Cap analysis graphs (Fig. 4). Usually pairwise comparisons are listed in a table only when no graphs (MDS or CAP analysis graphs) are present in the paper; when such figures are presented, instead, only significant pair-wise comparisons are described in the text (please see for example Smale et al., 2011; Engelen et al., 213; Guarnieri et al., 2014). Anyway, we can add a list of them if the Editor believes they are important.

I understand the authors’ comments here. I have address the presentation issue of Table 2 in comment 3. I believe it is fine to focus on the significant pairwise comparisons.

I’d be interested in seeing a figure like Fig. 5 for T2, we know that these communities are much more similar to each other, but we don’t know what their composition looks like.

Additionally, I agree with reviewer 2’s suggestion about how to treat the community composition data with regards to grazing. I believe it’s important to compare changes between T0 to T1 and T1 to T2 in order to treat the grazing variable appropriately. When treated this way, the response variable will show larger long-term differences for the 100% grazing treatment as it seems these will be the communities to have changed the most over your long-term time period (T2).

2. It would be useful to have a table with the SIMPER results to show the percentages of the differences in assemblage associated with the dark filamentous algae, encrusting coralline algae, and macroalgae (lines 200-204).
Following the suggestions of Rev. 2 the SIMPER analysis was not presented in the ms and a CAP analysis was instead used to visually represent each treatment and simultaneously examine the relative contribution of the dominant benthic taxa to these differences (Fig. 4 and L. 235-238, 259-262, 276-77.

I agree with reviewer 2’s suggestion. Thank you for addressing this comment.

3. The discussion in lines 214-217 on the long-term macroalgal response to disturbance needs more explanation. It is unclear that Table 2 has to do with the second sampling effort at T2 (line 217).
We are sorry but we don’t agree with the referee. We think that the information contained in lines 214- 217 (Paragraph 3.2) is complete as this short paragraph only reports the results we obtained on the long-term macroalgal response to disturbance and not the discussion on this part of the study, which is conducted in the discussion paragraph (Paragraph 4). Furthermore, in Table 2 only the results of the statistical analysis regarding the first part of the study (short-term macroalgal response) are presented, as it is clearly indicated in the caption. The results of the long-term response are instead reported in Table 3. Anyway, some more information was added in the above mentioned paragraph (L. 273-277).

The table captions are now clearer. It maybe be useful to divide the 2 tables that make up Table 2 into separate tables for clarity.

4. The discussion, with some grammatical and English language clean-up, does a good job of explaining your results in the context of some of the literature. If you could expand on the discussion of the importance or implications of the findings as it pertains to MPAs, the field of study, or management in the conclusion section, this would strengthen this section and make it relevant to a broader audience.
DONE: Some considerations on the suggested topic were added in the conclusion section (L. 367-372; 382-387).

I commend the authors on the re-writing of the discussion. All of my comments were addressed appropriately.

5. It is important to include a discussion of the limitations of the study in the discussion section (eg. small sample size, seasonal variance, etc.).
DONE: We added a short paragraph regarding the limitations of the study was added at the discussion (L.373-381).

Thank you for addressing this comment.

6. In the text of the results please discuss the quantitative enrichment of N and P in the nutrient treatments. N was enriched about 2 times the concentration of the water around the non-enriched plots, while P was enriched much more. Discuss how this may have affected your results, if at all.
All the fertilizer we could used had similar proportions in NP, as in agriculture these were the most productive concentrations. As in the sea limitation by P is more common and the situation present in plots allowed us to exclude such limitation, we believe no problems are present.

Thank you for addressing this comment.

Comments for the author:

7. I suggest the authors get editing help from someone with full professional proficiency in English.
DONE: An English revision was done by a mother tongue.

I commend the authors for getting editing help for the English grammar. The manuscript has highly improved because of this and it was much easier to read. It was clear and unambiguous. There are a few very slight grammatical errors here and there but they did not impede me from understanding what the authors were trying to convey.

8. A more in-depth discussion of the results is required with accompanying evidence in the form of relevant figures and tables.
DONE: the discussion section was improved and relevant figures and tables were cited in it (L. 282, 285, 299, 302, 323, 328, 342).

Thank you for addressing this comment.

Additional comments

To the authors - thank you for taking the time to address the reviews so thoroughly. I enjoyed re-reading your newly improved manuscript. It is much more clearly written now. I believe that with some minor revisions, it will be ready for acceptance.

Reviewer 2 ·

Basic reporting

The revised manuscript by Caronni et al. is much improved from its initial submission. The authors have made a good effort to make the introduction and methods more informative and to improve the presentation of their results. However, the quality of written English remains unsuitable for publication as there are numerous grammatical and structural errors that lead to confusion. I have provided a few examples of these in the introduction but have not highlighted them throughout the rest of the manuscript.

I still think more work is required to honestly and transparently portray the most meaningful results of the study. At present, the reader has to work far too hard to unpack the valuable pieces of information. See my comments and suggestions below in the validity of findings section.

Line 56: the word also is misplaced here. This sentence is also very confusing.

The point is that erect macroalgal taxa (Dictyota and Laurencia) were negatively impacted by grazing and had the highest abundance in plots that were mucilage-free, enriched in nutrients, and had limited grazing. Correct? I suggest rewording to make that more clear.

Lines 59-60: I do not understand this sentence. This is the end of the abstract, you should distill a clear, simply, take-home message for the reader.

Line 91: lead to regime shifts nearly irreversible

Line 92: been therefore done needs to be changed to “therefore been done”

Line 97: the word stressor appears to be missing after the word multiple

Lines 103-109: this section needs to be improved for clarity. It currently reads as a series of vague terms and big concepts. The authors should use examples from these or other references to elucidate how this information relates to their study and study system.

It is more important to help the reader understand why you are discussing these ideas than it is to simply reference them because this is a study on cumulative impacts.

Line 113: Ritson-Williams et al. 2009 focuses explicitly on CCA and its effect on the recruitment of tropical reef corals. As cited, this reference is used to discuss “canopy macroalgae,” which, if anything, have a directly negative impact on coral recruitment. Coral settlement is also not relevant for this study.

Line 138: herbivore intensity should be intensity of herbivory

Line 168: fast proliferate should be proliferate rapidly

Lines 177-180: The use of the term strengthened here is confusing. I think you mean that the negative impacts of the mucilage on the macroalgal communities can be strengthend by currents…because the currents move the mucilage around and it can wrap around and remove/scour other algal communities.

Please reword to clarify your meaning.

Lines 213-215: there are words missing here.

Experimental design

The methods read much more clearly in this version of the manuscript. I thank the authors for their attention to detail and for providing additional text and figures to clarify their approach.

Major comment:
- I think your grazing treatment still needs to be considered a little more thoughtfully. To be honest, I’m not convinced it accurately simulates “grazing.” By scrubbing the substrate with an iron brush you likely removed far more material than actual grazers would. This is why the effects of the grazing treatment overwhelm your other results.
- Specifically, how was the 50% grazing treatment enacted? Did you fully scrape 50% of the plot or did you trim 50% of the biomass? Was the scrapped part of the plot a contiguous half? Or where the scrapped areas distributed randomly throughout?
- Explaining that you used this method to approximate grazing of the dominant herbivore in your system (i.e., urchins would be helpful). Also briefly describing what type of urchins they are and what their grazing is like (are the boring urchins that don’t move much spatially and have very small grazing halos, or are they large roaming urchins that can eat everything?)

Minor comments:

- Need to indicate somewhere that these analyses were conducted in PRIMER +Permanova
- Also, there should be tests for normality to satisfy the assumptions of ANOVA

Validity of the findings

The biggest finding of this manuscript, in my opinion, is that the effects of the treatments are effectively gone by time point 2. The authors have discussed this briefly at the very end of the discussion but I think it deserves more attention. Notably, what was the biggest change in the plots between the two time points? What specifically, changed to make their multivariate composition identical?

All of the algal taxa examined here have very different growth rates and recruitment strategies. It is very possible that a bloom of DFA in all plots that had any free space (grazing treatment) overwhelmed the effect of treatment by T2. Data to address this and a more in depth natural history discussion on how the patterns in algal communities observed match known life histories (i.e., turf grows fast and recruits constantly while CCA is much more slow growing) would add more value to the manuscript.

With regards to my comment on grazing above I think the discussion requires some more attention to how the method of “grazing” could be biasing the interpretation of the results. Specifically, the results of all other treatments are minimal compared to grazing (fig. 5). At most taxonomic vary by a few % between mucilage and nutrient treatments when grazing was 0%. This is most likely just noise in the data from the coarse percent estimates used in the photographic analysis.

Additional comments

Comments to the author
In revision please pay particular attention to the use of grammar and sentence structure. Additional revisions by a native English speaker will go a long way to improve clarity of the manuscript.

Please also make an effort to distill the specific findings of your results rather than referring to general trends (e.g., Lines 359-361 – “significant differences were recorded”…what are those differences?) This will make your paper far more useful to managers as indicated in your discussion. As written it is still very challenging to extract the core take home messages and key results.

---

## Round 0.4 · Major Revisions

I asked one of the two previous reviewers to do a third review on the manuscript. The reviewer worked very hard on their review, and I expect your revision will attend to the specific issues raised, including the failure to include my recommended changes to Figure 4 in the text. The reviewer also expressed both in the review and privately to me continued frustration with the poor English grammar. Neither the reviewer nor I think it is fair that English is the lingua franca of science, but it is our responsibility to ensure that manuscripts are clearly written in English for the international audience of PeerJ. While you have improved the grammar extensively, the reviewer and I agree that many parts remain difficult to interpret correctly and a native english speaker, with a scientific background, should carefully revise the manuscript. Please make every effort to do so.

Reviewer 2 ·

Basic reporting

This is the third version of this manuscript that I have reviewed. While the authors have made improvements to the original draft, multiple errors remain that must be revised prior to this manuscript being considered acceptable for publication. There have been numerous requests by the editor and reviewers to have this manuscript evaluated and revised by a native English speaker. While some of the manuscript has been improved there are clearly sections (towards the end of the discussion for instance) that have not been addressed and therefore confound the interpretation of the manuscripts and its data.

There are also numerous grammatical and punctuation errors throughout the manuscript that should be cleaned up prior to subsequent submission.

Perhaps most importantly, the authors revised figure 4 based on the comments from the editor but did not update any of the text pertaining to this figure or the statistical methodology in the methods section. It is critical that the authors update their statistical methods, revise their data interpretation in the results section, and ensure that their conclusions match the new figures and analysis that they conducted. Without this, the paper is not suitable for publication.


Some line by line comments are provided below and should aid in the author's revision of the manuscript for clarity and accuracy.

Abstract

Grammar issues in the last two sentences

1) Sentence beginning With regard to…
a. It is difficult to follow the meaning of this but if I am understanding it correctly, I would suggest rewording to simply,

“We did not observe long-term effects of the treatments on the overall assemblage of the macroalgal community at our study site”

2) The last sentence should read something more along the lines of,

“These results illustrate the capacity of a shallow-water macroalgal community to quickly
recover from the simultaneous impacts of herbivory, nutrient enrichment, and mucilage.”


Main Text
Please make an effort to proof read the manuscript for typos, capitalization errors, and punctuation as they are numerous instances that are beyond the scope of this review to address.

Line 72 – shifts in what? This should be clarified.

Lines 85-86 – watch tense here. For both a and b “are” should be “is” if the tern effect is used. If you want to refer to several effects than keep “are” and make effect plural.

Lines 87-89 – As written this example is not very clear. Is the point that heavy metal toxicity increases under lower pH...because the metals become more biologically available? I would rephrase this and make sure to explain to the reader more thoroughly so that it helps highlight your point.

Lines 89-91 – This example is also confusing as written. It would be useful to explain this more explicitly to the reader…are low temperatures stressful? Or is that high temperatures? And what food levels interacted with temperature changes to drive the change in reproductive capacity?

Consider distilling both of these examples to their key points more directly. Perhaps something like, “(synergistic or antagonistic?) Interactions between temperature and food availability can reduce/enhance reproductive capacity in Daphnia spp.”

Lines 92-97 – I don’t understand this paragraph and I don’t think it is necessary to the introduction. I would remove it or revise to make the point much clearer.

Line 101 – This is still not a correct use of this citation. In the Ritson-Williams paper there is no direct mention of any “canopy macroalgae” or that inverts settle on macroalgae. There is one photo that illustrates a coral larva that settled on a piece of Ulva (non-canopy forming) but the context of this observation is negative and contrary to the hypothesis that you put forth in this sentence. If the facilitation of larval settlement is a point that you would like to make then you need to include a stronger and more accurate reference. If not, I this statement should be removed entirely.

Line 110 – Please use consistent terminology for this process throughout the manuscript. I would suggest sticking with “nutrient enrichment.”

Lines 111-115 – This sentence should be re-worded for clarity and I think the bit about regime shifts should be remove or described more thoroughly in another sentence. There are also several grammatical errors and missing punctuation. An example of how this sentence might be improved is offered below.

Also, the role of nutrient enrichment in determining macroalgal abundance has been extensively debated (refs). Recently, it has been suggested that nutrient enrichment may mitigate over-grazing by enhancing algal growth.

Lines 115 – 117 – This sentence appears to be unrelated to the preceding sentence. I would suggest removing it entirely and beginning a new paragraph with Line 117.

Line 117 – As this sentence begins a new topic it is helpful to start a new paragraph. I would suggest revising this sentence to be a better topic sentence. For example, “Most studies focus on the role of nutrient enrichment on macroalgal communities but it is also an important driver of microalgal proliferation.”

Line 131 – As written this sentence is confusing. What are the response variables? Concur should be “occur”. I would suggest revising this sentence to be more specific. For example,

Excess mucilage can have detrimental effects macroalgae and other benthic organisms and should therefore be considered an emerging threat to coastal ecosystems.


Lines 141-145 – This paragraph comes out of nowhere and does not real constitute a paragraph on its own. I would link this more directly with the preceding paragraph on mucilage and our limited understanding on its impacts on benthic organisms.

Line 160 – you don’t need the word “completely” here or in line 192.

Line 200 – what is N036? Do you mean N=36?

Statistical analyses – The authors reported to the editor that figure 4 now represents an NMDs plot rather than a CAP but the methods appear to not have been updated since the previous version of the manuscript. The authors must ensure that all statistical analyses, their assumptions, and the relevant procedural information is supplied in full in the methods section. And this information must match what is displayed in the figures.

Line 248 – what is (GMAV5)?

Line 296-297 – This conclusion appears contrary to your results, which showed that nutrient enrichment only had an effect in the presence of other stressors [lines 260-262] (e.g., grazing). I would stress that throughout the discussion care must be taken to ensure that your conclusions match the data you present and to be cautious with the over generalization of your results.

Lines 303-305 – This sentence doesn’t make sense. Are you saying that nutrient enrichment didn’t impact macroalgal assemblages in the presence of additional stressors? This also seems contrary to the results explained in lines 260-262, but perhaps I am missing the point because the phrasing is difficult to interpret.

Lines 350-352 – This appears to contradict the conclusion in the above paragraphs that mucilage mitigates the effect of nutrient enrichment on macroalgal communities because it takes up the excess nutrients much faster. After reading this statement, the reader is thus likely to be confused when they see a conclusion that nutrient enrichment can enhance macroalgal growth and “nullify” the effects of mucilage.

I suspect there is some misinterpretation here on my part but this underscores the need for the authors to yet again spend time clarifying their writing. The sentence structure remains convoluted and the certain terminology may be misleading.

Perhaps the point of this sentence is to highlight that there is no negative physical effects (e.g., erosion or mortality) of macroalgae in a nutrients + mucilage plot? But in that case, there are no data presented to support that statement.

Lines 364 – 373 this paragraph should be re-written with the assistance of a native English speaker. The authors note that they have had help in revision but the phrasing of lines 368-369 provide clear evidence that a more thorough revision is required,

Lines 368-369 - “it will remain unknown, as no longer data are unfortunately available.”

Lines 375 – 383 – This conclusion would be much more informative if it was written in the context of the ecological response rather than the vague synergistic/antagonistic effects of the stressors on each other. Remind the reader what the key biological response of the study taxa were in each of these respective treatments. Conclude what changes would be expected at a larger scale under the presence of these different stressor scenarios.

Lines 384 – 395 – This paragraph offers the reader no tangible information about the study. Notably, it is mentioned that useful guidelines are provided to managers but there is no place in the discussion where these are made explicitly. The conclusion offers the authors an opportunity to clarify their findings and examine them in a broader context but I would encourage them to actually reiterate their results here rather than use generic terminology that provides no concrete path forward for managers interested the issue of mucilage on their reefs.

Experimental design

This has been addressed in previous reviews. I have several minor comments and questions in my line by line comments that are attached.

Validity of the findings

I still think that some of the conclusions to this study are over-generalized or overstated. Additionally, based on the new figure 4 I would argue that the authors could make some stronger conclusions about the interaction between mucilage and nutrient enrichment, however, given that they have not updated their interpretation of this data in the main text I have not provided much detail on this aspect in my revision.

Additional comments

I understand the review process can be arduous at times and I appreciate the desire to complete this project and have the manuscript published. However, I strongly encourage you to take time with your revisions of the next version of this manuscript. Improving the writing with the advice of a professional editing service is something worth considering as the manuscript remains difficult to follow despite several attempts to improve it. Additionally, you updated statistical analyses and generated a new figure but did not revise the text in your methods section or results. That was a critical oversight and prevented me from providing this manuscript with a complete review.

---

## Round 0.5 · Major Revisions

Because this manuscript has been through so many revisions with Peer Reviewers, I have endeavored to do a final and full Editor review myself. Below are my broad and specific recommendations for improvement to be considered in a final revision before acceptance.

Figure 2 is misleading. Figure 2 should be removed and instead made into a third panel to accompany the NMDS in current Figure 4 (New Panel 4A). The figure should be rebuilt and should fully organize the experiment (all 12 treatments) and show the symbols used in Figure 4. This figure should work as a legend for Figure 4, detailing how each treatment is separated into symbol shapes (circles, diamonds, triangles for Grazing treatments), symbol fills (closed and open for nutrient enrichment) and symbol colors (black or grey for mucilage removal treatment). Please replace “Mucilage” with “Mucilage Removal” and please replace “Grazing” with “Grazing Exclusion” to clarify the actual treatment manipulation. I still feel that the G, E and M terms make sense, so don’t change those.

Figure 3 should be presented as a table with n, means and standard deviations. Statistics should be used to test which of the treatments differ significantly for each nutrient at each timepoint: A simple set of 4 t-tests comparing E+ and E-, one run for N and one for P at T1, another run for N and one for P at T2. The timepoints should be relabeled to indicate the duration from the start of the experiment. In this case I believe this is roughly 3 and 6 weeks from the initiation of the nutrient amendments. This is critical because T1 and T2 mean different things for nutrient sampling and biomass sampling.

Figure 4 needs a legend as a separate panel, which will be made from Figure 2 as noted above.

Figure 4 panels A and B will be changed to B and C, and should be labeled on the figure with the timepoints. The timepoints should be renamed to the actual durations from the start of the experiment. In this case

Figure 4A: grey diamonds are all solid, but half should be open diamonds.

Figure 5: Clarify what timepoint (duration from start) of the experiment is shown here.
Table 4 can be removed and just noted on the NMDS (Current Figure 4B) that PERMANOVA on T2 was not significant for any term.

All figures and tables and throughout text: Rename ECA to CCA (Crustose Coralline Algae) as is standard in the literature.

L48-50: Sentence starting “According to different treatments” should be replaced with:
“Factorial combinations of 3 experimental treatments were applied in triplicate, including three grazing levels crossed with two nutrient enrichment and two mucilage removal treatments”

L50: Remove paragraph break. Abstract should be 1 paragraph.

L51: Replace “at the short term” with a more specific duration.

L52: “in enriched plots” please clarify which experimental treatment: nutrient enrichment alone?

L166: Remove the reference to personal communication with the lead author. Instead report mean ambient concentrations of N and P measured in this study.

L167: replace “graziers” with “grazers”

L171: replace “Their” with “Urchin”

L173: How were the “two rocky areas” accounted for statistically? Was one fertilized and one not? Were all treatment combinations randomized across both areas? (See confusion again in Line 194)

L176: replace “of the hereafter described treatments” with “treatment”

L177: “three factors: three grazing levels crossed with two nutrient enrichment and two mucilage removal treatments, following a factorial design and three replicates…”

L181: Clarify when the scraping of the substratum occurred in the timeline.

L221: Please do a square root transform on these data before generation of Bray-Curtis matrix input to PERMANOVA because they are relative abundance (% cover).

---

## Round 0.6 · Minor Revisions

Thank you for your attention to the lengthy suggestion and revision process with our excellent hardworking peer reviewers!

The science is now acceptable, however, the Section Editor who reviewed this decision has indicated that the English is still not yet acceptable. They feel that the authors really need to seek the help of a professional language editing service to improve the English language before Acceptance.

---

## Round 0.7 · accepted · Accept

The manuscript is much improved. Thank you.

#